# Are All Societies Ready for Digital Tools? Feasibility Study on the Use of Mobile Application in Polish Early Breast Cancer Patients Treated with Perioperative Chemotherapy

**DOI:** 10.3390/healthcare11142114

**Published:** 2023-07-24

**Authors:** Grażyna Suchodolska, Anna Koelmer, Monika Puchowska, Elżbieta Senkus

**Affiliations:** 1Department of Oncology & Radiotherapy, Faculty of Medicine, Medical University of Gdańsk, Smoluchowskiego 17, 80-214 Gdańsk, Poland; gsuchodolska@gumed.edu.pl; 2Centre of Biostatistics and Bioinformatics Analysis, Medical University of Gdańsk, Dębinki 1, 80-211 Gdańsk, Poland; anna.koelmer@gumed.edu.pl; 3Department of Non-Commercial Clinical Research, Clinical Research Support Centre, Medical University of Gdańsk, Skłodowskiej-Curie 3a, 80-210 Gdańsk, Poland; monika.puchowska@gumed.edu.pl

**Keywords:** chemotherapy, early breast cancer, eHealth, mobile app, symptom management, symptom reporting, ePROM

## Abstract

Background: The population of individuals affected by breast cancer is growing, and with advances in cancer treatment implemented into usual care, there is an urgent need to improve the recognition, monitoring and treatment of therapy-induced adverse effects. This study aims to explore the use of an in-app electronic questionnaire to assess and monitor chemotherapy-related symptoms in early breast cancer patients treated with perioperative chemotherapy. Method: Between December 2019 and June 2021, 72 female study participants used the mobile app *Centrum Chorób Piersi UCK* and completed an in-app questionnaire about the 14 most common chemotherapy-related symptoms. Replies including symptoms with a critical value triggered automatic email alerts to the nursing team. Results: Acceptance of the study was higher among younger women and patients originating from rural areas, while possible digital exclusion among patients >60 years was observed during the enrolment process. A total of 55 participants completed the electronic questionnaire at least once and generated 553 responses with 1808 specific problems reported. Fatigue (n = 428) was the most common problem, and fever (n = 5) the least reported problem. A total of 21 participants triggered alerts with responses containing symptoms with critical value assessment (n = 89). Significant negative correlation was observed between the number of responses and time from the first chemotherapy administration; however, the number of responses was not determined by any sociodemographic or medical factors. Significant positive correlations were identified between the number of communicated problems and participants’ age. The usage of our electronic symptom assessment questionnaire decreased substantially after the period of active encouragement during the study enrolment. Conclusions: Not all societies are ready for innovative eHealth solutions. Patients’ age should be carefully considered when app-based interventions are introduced to usual cancer care. Additional support is suggested for older patients to improve their awareness and participation in eHealth interventions. More research involving older participants is needed to explore and address their particular needs and perspectives on eHealth solutions.

## 1. Introduction

Breast cancer is the most prevalent cancer diagnosis and the second most frequent cause of cancer-related deaths among Polish women. According to the Polish National Cancer Registry in 2020, 23.8% of primary cancer diagnoses in women were breast cancers. While the population of people diagnosed with breast cancer in Poland is growing [1] and advances in cancer treatment are being implemented into usual care [2], there is an urgent need to improve the recognition, monitoring and treatment of therapy-induced adverse effects. Electronic patient reported outcome measures (ePROMs) such as mobile and web applications (apps) offer the opportunity to address particular unmet needs of cancer patients. They can be utilised for screening, diagnostic, therapeutic and educational purposes [3]. Various in-app questionnaires facilitate capturing patient-reported outcomes, such as symptom burden, physical function, mental status and quality of life [4,5,6,7,8]. Implementing electronic monitoring of treatment side effects into routine breast cancer care decreases the chemotherapy-induced symptom burden, and improves symptom management, self-efficacy and quality of life [8,9,10,11,12,13]. Moreover, several studies exploring mobile apps encouraging physical activity among breast cancer patients demonstrated improved quality of life and reduced fatigue and distress levels [14,15,16]. Additionally, the results of research testing eHealth solutions during the COVID-19 pandemic not only confirmed that remote monitoring and management of treatment-related symptoms reduce the symptom burden and improve the quality of life, but also limit unplanned healthcare utilisation, decreasing demand on healthcare systems and improve patients’ symptom experience, including perception of the frequency, intensity, distress, and meaning of symptoms [6]. Furthermore, eHealth solutions have been identified as empowering, improving patients’ involvement in the continuum of cancer care [17,18].

The primary objective of this study was to analyse the results of using an in-app electronic questionnaire to assess and monitor chemotherapy-related symptoms in patients treated for early-stage breast cancer.

## 2. Materials and Methods

### 2.1. Study Design and Participants

This study assessed the use of a mobile app used by Polish early breast cancer patients treated with perioperative chemotherapy. The study was conducted in the Breast Cancer Chemotherapy Day Unit of the University Clinical Centre (UCC) in Gdańsk, Poland, between December 2019 and June 2021. Approval from the Independent Bioethics Committee for Scientific Research at the Medical University of Gdańsk (NKBBN/642/2019, NKBBN/642-534/2020) was obtained before the study initiation. 

Patients who met the following inclusion criteria were eligible for participation in this study: referral for perioperative chemotherapy for early-stage breast cancer (all types), possession of their own smart device, and an ability to navigate the device and download mobile applications independently; these patients completed a signed informed consent form. Patients with metastatic breast cancer and those treated for other types of cancers were excluded from the study. 

Perioperative chemotherapy was categorised as neoadjuvant or adjuvant with further division into specific treatment regimens (Table 1). Anthracycline-taxane regimens included combinations of doxorubicin and cyclophosphamide with paclitaxel +/− carboplatin or monoclonal antibodies (trastuzumab, pertuzumab), whereas anthracycline or taxane-based chemotherapy consisted of doxorubicin or docetaxel combined with cyclophosphamide, or combinations of paclitaxel, carboplatin and monoclonal antibodies. 

### 2.2. Measures

During the pre-treatment preparation process, participants were asked to download the free mobile app *Centrum Chorób Piersi UCK*, which contains an electronic questionnaire (ePROM) for monitoring chemotherapy-related adverse effects. Nursing staff collected medical data (cancer stage and phenotype, setting of chemotherapy (preoperative vs. postoperative), type of cytotoxic drugs used, number of cycles, coexisting medical conditions and use of granulocyte colony-stimulating factors) from patients’ health records. Sociodemographic details (age, sex, place of residence, education, employment, economic and marital status) were self-reported by participants at the end of the patient satisfaction survey. 

Enrolled patients were instructed by the breast cancer nurse (BCN) to complete the questionnaire weekly and on occasions when they experienced distressing symptoms. To ensure participants’ safety, it was emphasised that in the case of an emergency or sudden health deterioration, they must seek medical help through emergency services or an appointment with a physician or nurse, as appropriate. Similar information was displayed on the questionnaire’s summary screen before submission. 

The questionnaire consists of 14 questions about the most common chemotherapy-related adverse events (Appendix A). The symptom severity assessment scale used in the questionnaire is based on the Common Terminology Criteria for Adverse Events v.4 [19] with its own modifications. Participants defined relevant symptoms on a 5-point scale, where 0 meant no problem at all, 1—mild, 2—moderate, 3—severe and 4—debilitating problem. The application did not allow omitted questions or free-text responses. Reports generated in connection to patients’ in-app activity were closely monitored by the BCN. Figure 1 demonstrates a simplified patient–breast care team communication process via the in-app questionnaire.

Replies including symptoms with critical value (≥3, apart from fever, which activated alerts when rated as 1 or above) triggered automatic email alerts to the nursing team. Participants were informed that after triggering an alert they will be contacted by the BCN within one working day during office hours. Nursing interventions performed in response to app alerts were recorded in patients’ documentation. Every intervention began with a telephone consultation leading to further advice as necessary, and the types and outcomes of the recommended interventions were analysed. Additionally, to evaluate patients’ opinions on the application, including the electronic questionnaire, participants were asked to complete a paper survey (Appendix B) after finishing the last chemotherapy cycle or after the physician’s decision to terminate the treatment. The proprietary survey consisted of 10 questions relating to: the difficulty level of using the in-app questionnaire, safety during treatment, satisfaction from received care, sense of control, well-being during treatment, hospital admissions and free text option for additional participant suggestions.

### 2.3. Data Analysis

Data were managed with Microsoft Excel software and statistical analyses were performed with free RStudio software, version 4.2.2 (R Core Team, 2021) under the terms of the General Public License. Medians were used to report the central tendency, as the data were not normally distributed. Nonparametric statistical methods were used to analyse the results of this study due to the small population and non-normally distributed outcome data. An α level of 0.05 was set for all tests. Correlations were assessed with the ρ-Spearman’s correlation coefficient and its significance test. The Mann–Whitney U test was used to describe differences in continuous variables between two groups and the Kruskal–Wallis test for multiple (>2) groups. Post hoc Wilcoxon tests were performed to determine which groups were significantly different. For comparison between two categorical variables, Chi-square tests (for values > 5) and Exact Fisher’s test (for values < 5) were used. Additionally, Wilcoxon Rank sum tests were performed to check differences between the median number of reported problems and two selected qualitative characteristics. Benjamini–Hochberg *p*-value correction for multiple comparisons was used for all performed tests. 

## 3. Results

### 3.1. Enrolment 

Between December 2019 and December 2020, 93 early-stage breast cancer patients were assessed for eligibility. A total of 20 women refused, and a single eligible man was excluded from the analysis by the research team, resulting in 72 women being included in the study. 

Acceptance of the study was higher among younger women and patients originating from rural areas (Figure 2A,B). Although the reasons for refusal were not systematically collected, the most frequently observed explanations included difficulties with navigating through mobile apps, no access to a smartphone or limited ability to use it.

### 3.2. Participants’ Characteristics

Medical and sociodemographic characteristics of study participants are presented in the following table (Table 2).

Medical conditions reported by participants included: hypothyroidism (n = 20), hypertension (n = 7), asthma (n = 4), endometriosis (n = 3), obesity (n = 3), varicose veins (n = 3), coronary disease (n = 1), irritable bowel syndrome (n = 1), glaucoma (n = 1), nephrolithiasis (n = 1), epilepsy (n = 1) and rheumatoid arthritis (n = 1). Details of concomitant medication were not collected for the purpose of this study.

### 3.3. Participant Engagement with the App and Questionnaire

Data collection lasted from December 2019 to June 2021. During the study, 55 (76%) participants completed the electronic questionnaire at least once and of those 23 (42%)–10 or more times. Patients who used the electronic questionnaire and those who submitted no responses did not differ in terms of sociodemographic or medical characteristics (not reported). The number of responses generated by those who completed the questionnaire ranged from 1 to 68, median 7 (Figure 3). 

Only 6 (11%) participants filled in the questionnaire as instructed by the BCN; others chose to complete it only when experiencing problematic side-effects. Overall, 553 responses were collected and some referred to more than one symptom, resulting in 1808 reported problems (Figure 4A). Fatigue (n = 428) was generally the most frequently reported problem, followed by the sensation of pins and needles in hands and feet (n = 226), pain (n = 208) and dizziness (n = 196). The least common problems reported included vomiting (n = 6) and fever (n = 5). 1329 (73.5%) symptoms were mild, 392 (21.7%)—moderate, 80 (4.4%)—severe and 7 (0.4%)—debilitating. 

A total of 21 (29%) participants triggered alerts with responses containing symptoms with critical value assessment. Overall, 58 responses with values over the predefined critical threshold were collected, of which 21 (36%) referred to more than one symptom with a critical value (Figure 4B). Altogether, there were no statistically significant differences in sociodemographic or medical characteristics between patients who triggered alerts and the remaining participants.

Although there were no statistically significant differences in the number of generated responses in relation to sociodemographic details such as age, place of residence, education level, employment, economic or marital status, and medical factors including breast cancer stage, chemotherapy setting, chemotherapy regimen, number of treatment cycles, coexisting medical conditions or use of G-CSFs, a statistically significant negative correlation was observed between the number of responses and time from the first chemotherapy administration (Figure 5). 

For the purposes of analysis of correlations identified between communicated problems and specific participant characteristics, problems were arranged into five categories: fatigue, dizziness, gastrointestinal problems (appetite loss, nausea, diarrhoea, constipation, oral mucositis, vomiting), respiratory problems (shortness of breath, cough) and others (pain, skin changes on hands and feet, sensation of pins and needles in hands and feet, fever). Detailed responses collected during the study and grouped accordingly are presented in Appendix A and Figure 6.

The participants’ age was found to be the only factor determining the number of reported problems. Statistically significant positive correlations observed between age and the number of reported specific problems are presented in Figure 6A–D.

Additionally, factors including employment status, marital status and stage of the breast cancer were initially found to be significant in the Kruskal–Wallis test; however, further analysis (Wilcoxon signed-rank tests) found these to be statistically insignificant (*p* > 0.05) (Appendix A). Separate sensitivity analyses (not reported) confirmed that results are not affected by the inclusion of a single male participant or the exclusion of participants who visibly exaggerated the number of reports (Figure 3). 

All participants completed the patient satisfaction survey. The majority (89%) of respondents assessed using the in-app questionnaire for symptoms reported it to be very easy to use. Moreover, most of them believed that the possibility of reporting symptoms via an electronic questionnaire not only improved their safety and well-being during treatment, but also made them feel in control of the situation in which they found themselves. Most participants felt satisfied with the support they received in response to the submitted forms. More than half of respondents thought that using the side-effects-reporting module improved the treatment of bothersome symptoms. According to the vast majority of participants, the overall care and support provided by medical staff met their expectations, with only a small percentage feeling that their expectations were not met. Ten participants expressed their suggestions of adding extra questions about nail problems (n = 4), cardiac complications (n = 1) and other unspecified unincluded symptoms (n = 5). Detailed responses to specific questions are presented in Table 3.

### 3.4. Nursing Interventions 

During the study, 58 nursing interventions (telephone consultations) were performed in response to app alerts generated by 21 participants. All interventions provided psychological support and reassurance combined with additional guidance as required. Topics covered during telephone consultations included: physical activity and sleep (74%), pain management (24%), drug compliance (16%) and dietary needs (10%). A total of 64% of nursing interventions related to single problem reports and 36% to alerts about multiple problems; mostly consisting of combinations of severe fatigue, pain, shortness of breath and dizziness. Overall, five (7%) patients in 23 interventions were advised to use emergency care services due to severe dyspnoea (n = 10), dizziness (n = 8), diarrhoea (n = 3) and fever (n = 2); however, no hospital admissions were required. The number of follow-up contacts was flexible and tailored to the patient’s needs; nevertheless, they were not analysed within this study.

### 3.5. Post-Study Observations

After the end of the study, we continued to provide new patients with information about the possibility of reporting chemotherapy-related symptoms via the mobile app. In the follow-up process of the study, mobile application reports were monitored for an additional year. Unexpectedly, the use of the mobile app for symptom reporting dropped substantially after the active study ended, resulting in only two patients using the in-app questionnaire to report treatment-related side effects over this period. 

## 4. Discussion

High acceptance of digital tools in cancer care is widely recognised [3,7,9,20,21,22]. Nevertheless, there are still societies that prefer the traditional way of communication. Specialist breast cancer nurse-led interventions, mostly telephone consultations, tailored to patients’ requirements are part of standard care for all breast cancer patients treated in UCC Gdańsk, where this study was conducted. Complex nursing interventions can significantly reduce the symptom burden [23,24], and symptom-focused education can enable patients to administer self-care more effectively [25,26]. With already well-established patient–nurse communication standards, the uptake of an additional tool to report treatment-related side effects by breast cancer patients was poor, especially after the period of active recruitment to the study. Moreover, possible digital exclusion among older (>60 yrs) patients was observed during the enrolment process, resulting in their lower interest in study participation (Figure 2), albeit the rationale for study refusal was not recorded. Similar digital inequalities between age groups were discussed in multiple studies involving participants with other diseases such as diabetes [27], irritable bowel syndrome [28], asthma and COPD [29], heart failure [30] and mental illnesses [31], and age was found to be a significant factor that influences the use of eHealth. However, although patients older than 60 years of age were generally less interested in participating in our study, once enrolled, they used the in-app questionnaire for symptoms reporting with comparable patterns of reporting, and overall, a pattern of reporting more problems than younger participants (Appendix A, Figure 6). 

Adherence to the *Centrum Chorób Piersi UCK* app decreased with individual progress throughout chemotherapy (Figure 5). Similar observations were made in another study in early breast cancer patients [32], where the authors suggested that the reason for dropping engagement was an improvement in understanding the nature of particular side effects and the development of coping strategies; therefore, patients did not feel reporting symptoms to still be necessary. App adherence in cancer care has been explored in another study [20], where predictors of adherence were evaluated; similarly to our findings, no significant correlations were identified. 

The prevalence of particular symptoms reported during our project is consistent with other studies. Fatigue is known to be the most frequent and distressing symptom for breast cancer patients [33,34,35,36]. In our study, fatigue accounted for the majority of overall problems reported, and triggers activating alerts to the BCNs. Dyspnoea was the second most common reason triggering alerts. Compared with another study testing a similar app solution [37], the average number of alerts per patient in our series was similar (2.8 vs. 2.7). However, in contrast to research led by Basch et al. [38], the proportion of individual symptoms triggering alerts in our study was higher (89/1808, 4.9% vs. 1431/84212, 1.7%), while the type of triggers remained consistent, fatigue, dyspnoea and pain being the most common severe or debilitating problems. Multiple studies examined factors related to fatigue; however, results are ambiguous. Several reports [39,40,41], found no significant relationship between fatigue and age. Other studies [42,43] have observed younger patients experiencing higher levels of fatigue, which was also associated with working while in treatment. In contrast, the present study suggests that older patients are at a higher risk of experiencing cancer-related fatigue. The breast cancer stage and number of chemotherapy cycles were insignificant in our investigation, while other studies [36,44,45] indicate that patients with more advanced nonmetastatic breast cancer and those receiving more treatments are at higher risk of suffering from cancer-related fatigue. The number of responses and specific problems communicated during our study were not determined by education level, marital, employment or economic status. Nevertheless, another study [21] demonstrated that married and cohabitating participants generated more reports than those living alone. The same study evaluated patients’ perceptions of using the app, and in line with the results of our study, demonstrate that the possibility of reporting treatment side effects in real-time created feelings of assurance and safety. Moreover, other research that explored the effect of eHealth apps on patient satisfaction during treatment confirmed that the use of mobile health apps could improve patient experience and overall health outcomes [22]. Recently published results of the PreCycle trial [46] present the successful utilisation of the interactive app (CANKADO), that works without any intervention by healthcare professionals to significantly delay the deterioration of quality of life among patients treated for metastatic breast cancer. The study reveals the next generation of ePRO monitoring and management, opening further discussion for patient empowerment and involvement in the continuum of cancer care.

### Limitations of the Study

The present study has some limitations. Our project was performed in a single centre with a limited number of participants due to the COVID-19 pandemic. To explore participants’ perception of the app, we used a proprietary survey, instead of a standardised tool; however, we achieved a 100% completion rate with overall positive feedback. Another potential drawback is that the study design did not involve automatic reminders for participants to complete the in-app questionnaire, resulting in low app adherence (e.g., only 6% of participants completed the questionnaire as instructed).

## 5. Conclusions

Although successful use of ePROMs for monitoring treatment-related adverse events has been described in many settings, the results of this study suggest a possible lack of trust and/or understanding of eHealth tools among Polish patients treated for early-stage breast cancer. Our findings suggest that patients older than 60 years of age find it difficult to engage with mobile technology and eHealth solutions. On the other hand, this is the population that, according to our research, is at a higher risk of experiencing not only cancer-related fatigue, but also other problems caused by the treatment. To improve patient engagement and understanding of eHealth solutions, it is essential that patients are invited to and involved in the fundamental stages of creating innovative app-based interventions. With age being a significant factor in determining the number of problems experienced during chemotherapy, we suggest that additional support be provided to older patients to enhance their awareness of the beneficiary potential of eHealth interventions. More research involving older participants is needed to explore and address their particular needs and perspectives on eHealth solutions.

## Figures and Tables

**Figure 1 healthcare-11-02114-f001:**
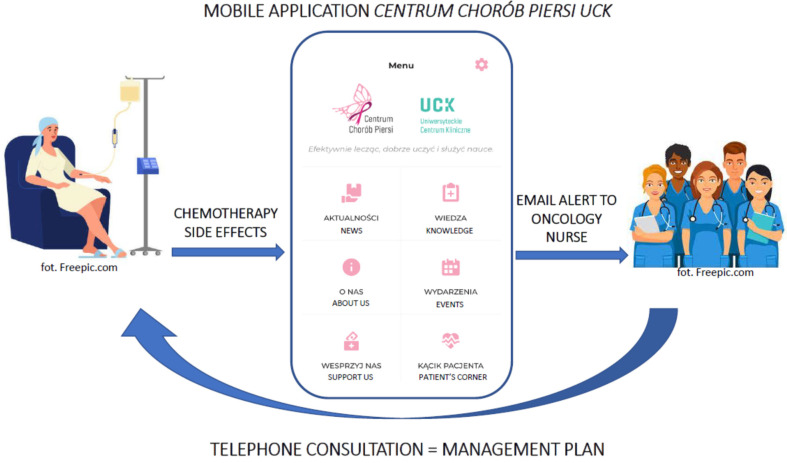
Patient–breast care team communication process via in-app questionnaire.

**Figure 2 healthcare-11-02114-f002:**
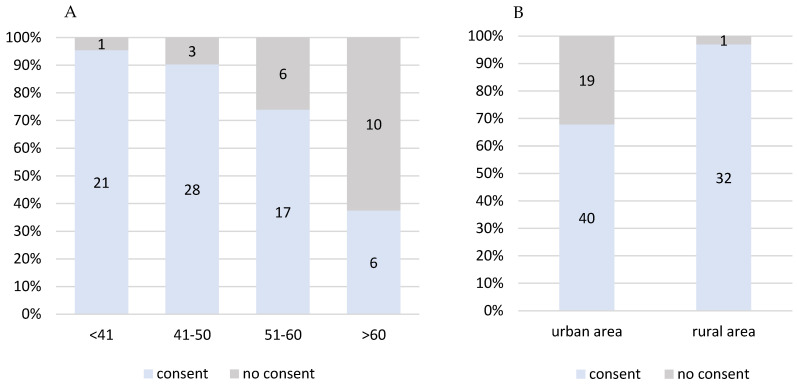
Study acceptance (**A**) in different age groups (*p* = 0.00006), (**B**) based on place of residence (*p* = 0.00114).

**Figure 3 healthcare-11-02114-f003:**
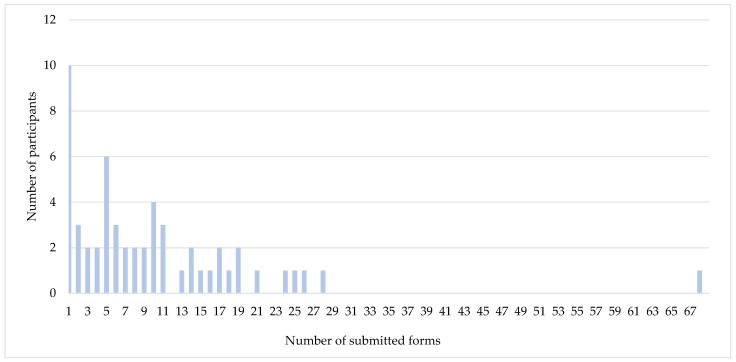
Responses generated during the study.

**Figure 4 healthcare-11-02114-f004:**
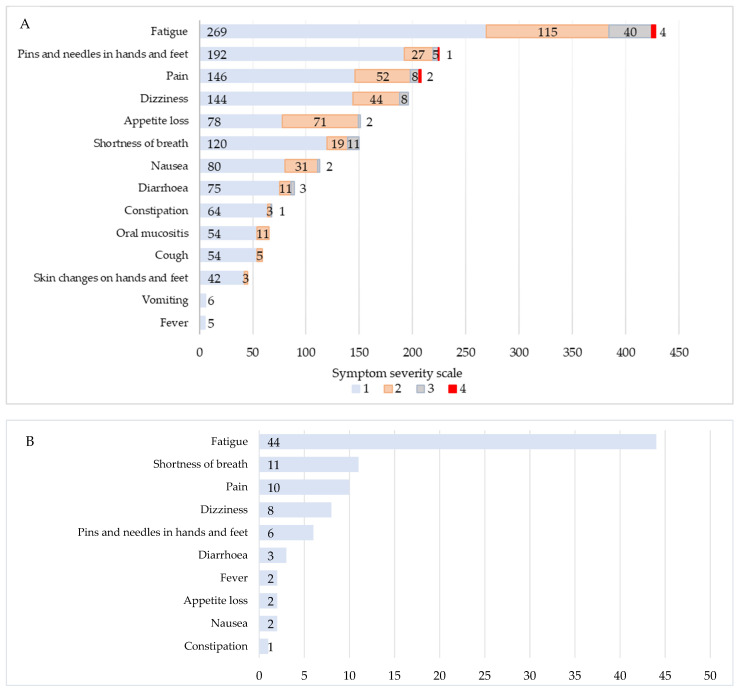
Summary of (**A**) specific symptoms reported during the study, (**B**) specific symptoms with critical value assessment.

**Figure 5 healthcare-11-02114-f005:**
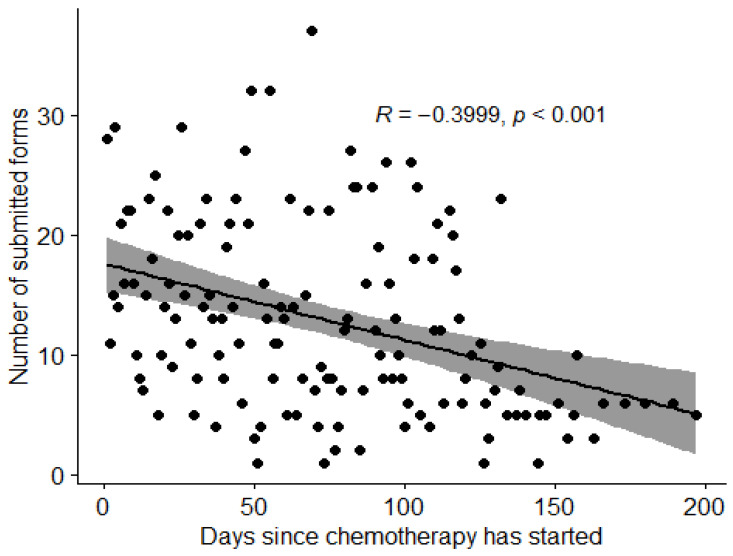
Distribution of responses collected during study period.

**Figure 6 healthcare-11-02114-f006:**
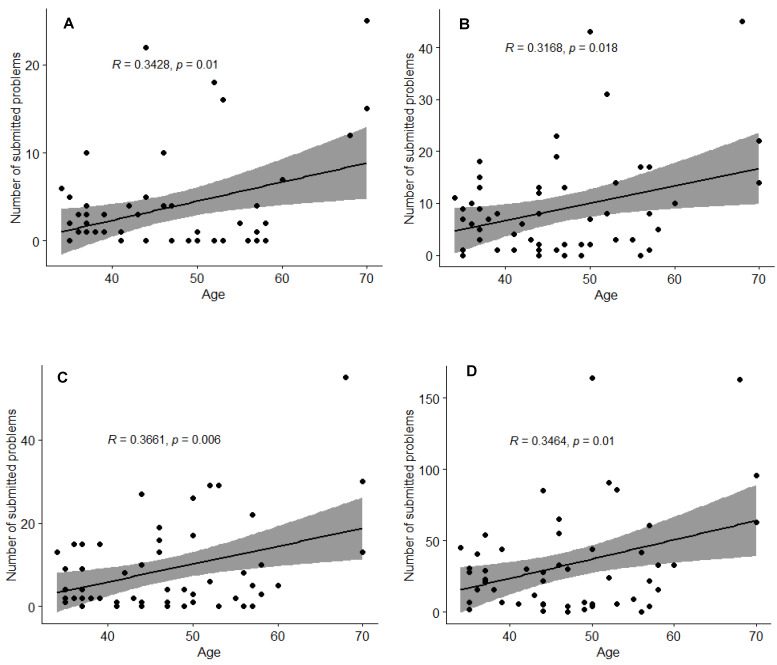
Correlations between age and (**A**) respiratory problems, (**B**) gastrointestinal problems, (**C**) other problems, (**D**) all problems reported.

**Table 1 healthcare-11-02114-t001:** Medical characteristics of study participants.

**Breast cancer type:**		
HER2+	21	29.2%
luminal A	12	16.6%
luminal B	19	26.4%
TNBC	20	27.8%
**Breast cancer stage:**		
IA	11	15.3%
IB	3	4.2%
IIA	27	37.5%
IIB	14	19.4%
IIIA	6	8.3%
IIIB	6	8.3%
IIIC	5	6.9%
**Chemotherapy setting:**		
neoadjuvant	53	74%
adjuvant	19	26%
**Chemotherapy regimen:**		
anthracycline + taxane-based:	58	80.6%
neoadjuvant:	48	66.7%
12xPXL>>4xAC	15	20.8%
12xPXL>>4xddAC	15	20.8%
12xPXL+carboplatin>>4xddAC	7	9.7%
4xddAC>>12xPXL + T	5	6.9%
4xAC>>12xPXL + T	6	8.3%
adjuvant:	10	13.9%
12xPXL>>4xAC	6	8.3%
12xPXL>>4xddAC	1	1.4%
4xAC>>12xPXL + T	1	1.4%
4xddAC>>12xPXL	1	1.4%
4xddAC>>12xPXL + T	1	1.4%
anthracycline- or taxane-based:	14	19.4%
neoadjuvant:	5	6.9%
12xPXL	1	1.4%
4xddAC	1	9.7%
12xPXL + carboplatin + T + P	3	4.2%
adjuvant:	9	12.5%
12xPXL + T	4	5.6%
4xTC	4	5.6%
12xPXL	1	1.4%
**Number of chemotherapy administrations:**		
16	56	77.8%
12	10	13.9%
7 *	1	1.4%
4	5	6.9%
**Treatment with G-CSF:**		
yes	33	46%
no	39	54%
**Coexisting medical conditions:**		
no	36	50%
yes:	36	50%
single condition	24	33.3%
multiple conditions	12	16.7%

* patient’s decision to change treating hospital. AC—doxorubicin-cyclophosphamide, dd—dose dense, G-CSF—granulocyte colony stimulating factors, P—pertuzumab, PXL—paclitaxel, T—trastuzumab, TC—docetaxel-cyclophosphamide, TNBC—triple negative breast cancer.

**Table 2 healthcare-11-02114-t002:** Sociodemographic characteristics of study participants.

Age	Median	Range
	46	(34–70)
	Frequency (n)	Percentage (%)
Total	72	100%
**Gender:**		
female	72	100%
male	0	0%
**Marital status:**		
married	52	72.2%
single	12	16.7%
divorced	3	4.2%
in relationship	3	4.2%
widow	2	2.8%
**Place of residence:**		
city > 50,000	27	37.5%
city < 50,000	14	19.4%
rural area	31	43.1%
**Education:**		
higher	39	54.2%
secondary	17	23.6%
vocational	16	22.2%
**Employment status:**		
employed	63	87.5%
unemployed	5	6.9%
retired	4	5.6%
**Economic status (self-assessed):**		
satisfactory	58	80.6%
unsatisfactory	14	19.4%

**Table 3 healthcare-11-02114-t003:** Detailed responses from patients’ satisfaction survey.

**1. In your opinion, how difficult it is to use the in-app side-effects-reporting module?**					
**Answer**	**Very easy**	**Rather easy**	**Difficult** **to say**	**Rather difficult**	**Very difficult**
N (%)	64 (89)	6 (8)	2 (3)	0	0
**2. In your opinion, did the possibility of using the in-app side-effects-reporting module improve your safety during chemotherapy?**					
Answer	Definitely yes	Rather yes	Difficult to say	Rather no	Definitely no
N (%)	44 (61)	26 (36)	2 (3)	0	0
**3. Are you satisfied with the support you received in response to submitted answers?**					
Answer	Definitely yes	Rather yes	Difficult to say	Rather no	Definitely no
N (%)	42 (58)	10 (14)	20 (28)	0	0
**4. Did you feel in control of the situation, when using the in-app side-effects-reporting module?**					
Answer	Definitely yes	Rather yes	Difficult to say	Rather no	Definitely no
N (%)	30 (42)	26 (36)	16 (22)	0	0
**5. In your opinion, did the possibility of using the in-app side effects-reporting-module improve your well-being during chemotherapy?**					
Answer	Definitely yes	Rather yes	Difficult to say	Rather no	Definitely no
N (%)	37 (51)	32 (45)	3 (4)	0	0
**6. In your opinion, did the use of the in-app side-effects-reporting module improve the treatment of bothersome side effects of chemotherapy?**					
Answer	Definitely yes	Rather yes	Difficult to say	Rather no	Definitely no
N (%)	22 (30)	23 (32)	26 (36)	1 (1)	0
**7. How do you assess the waiting time for the reaction of the medical staff in response to submitted answers about feeling unwell?**					
Answer	Very short	Rather short	Difficult to say	Rather long	Very long
N (%)	22 (30)	25 (35)	25 (35)	0	0
**8. Did the support provided to you by the medical staff meet your expectations?**					
Answer	Definitely yes	Rather yes	Difficult to say	Rather no	Definitely no
N (%)	56 (78)	11 (15)	3 (4)	2 (3)	0
**9. Did you require hospital admission due to exacerbation of chemotherapy-related side effects?**					
Answer	Yes	No			
N (%)	5 (7)	67 (93)			
**10. Is there anything that you would like to change in the in-app side-effects-reporting module?**					
Answer	Yes	No			
N (%)	10 (14)	62 (86)			

## Data Availability

The data presented in this study are available on request from the corresponding author.

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
