# Peer review of "Are All Societies Ready for Digital Tools? Feasibility Study on the Use of Mobile Application in Polish Early Breast Cancer Patients Treated with Perioperative Chemotherapy"

_healthcare, 2023, doi:10.3390/healthcare11142114_

Round 1

Reviewer 1 Report

Dear authors,

Thank you for your effort on the redaction of the current manuscript. It was such a pleasure reviewing your work.

The theme is original and deserves exploration in order to achieve data to assess an app usability.

This topic is adequate to the purpose of the Healthcare Journal, as it falls within this journal scope.

The title has a statement that seems to be a conclusion of the study, I suggest a reformulation in order to leave implies the question that is the main goal of the study.

The introduction refers “high acceptance of eHealth applications among cancer patients (early stage, metastatic and survivors) was identified by numerous studies”, this claim should be further explored. What kind of applications, what are their purposes? Were they all to manage symptoms related to drugs side effects?

The aim of the study is clearly defined.

In the Methodology section inclusion criteria should be clarified, indicating if all types of breast cancer were included in the current study. Also the medication used by patients, in addition to chemotherapy, should have been considered. The economic status classification should be clarified, regarding the tool used to this assessment by self report (I believe this is the way it was done).

Methods for statistical analysis are appropriate to the study design purposed.

In the Results section I leave the suggestion of split Table 1 in two tables, one describing sociodemographic characteristics and another presenting clinical data (including cytotoxic drugs). Also Fig.3 can be replaced by another figure, table or even descriptive text, it does not add any relevant information.

In the graph of Fig.4 it would be advisable for each bar to have the value of the respective frequency.

Table 2 is long and difficult to read, I suggest it be rearranged or included as supplementary material. Furthermore, there are no comments in the text regarding the contents of this table.

Information provided through Fig.7 is not easily understandable, this information includes the answers for questions Q1-Q8?? Please clarify the information provided in the legend or change the graph used. Later, questions presented in appendix A, the question number is not correspondent to those used in the Fig.7.

The Conclusions could be reformulated in order to present the study main achievements more clearly, since it does not address exactly the information passed in the results section.

The references are appropriate, up to date (more than 80% of the references were published in the last 10 years).

In general, the figures and graphs should be improved regarding their presentation.

In general, writing should be revised throughout the entire manuscript.

Author Response

Dear authors,

Thank you for your effort on the redaction of the current manuscript. It was such a pleasure reviewing your work.

The theme is original and deserves exploration in order to achieve data to assess an app usability.

This topic is adequate to the purpose of the Healthcare Journal, as it falls within this journal scope.

Thank you for this comment.

The title has a statement that seems to be a conclusion of the study, I suggest a reformulation in order to leave implies the question that is the main goal of the study.

Thank you for your suggestion We have rephrased the title to avoid the impression of a conclusion.

Are all societies ready for digital tools? Feasibility study on the use of mobile application in Polish early breast cancer patients treated with perioperative chemotherapy

The introduction refers “high acceptance of eHealth applications among cancer patients (early stage, metastatic and survivors) was identified by numerous studies”, this claim should be further explored. What kind of applications, what are their purposes? Were they all to manage symptoms related to drugs side effects?

Thank you for your valued comment. Referring also to other Reviewers, we have changed the introduction as follows:

Breast cancer is the most prevalent cancer diagnosis and the second most frequent cause of cancer-related deaths among Polish women. According to the Polish National Cancer Registry in 2020, 23.8% of primary cancer diagnoses in women were breast cancers. While the population of people diagnosed with breast cancer in Poland is growing [1] and advances in cancer treatment are being implemented into usual care [2], there is an urgent need to improve the recognition, monitoring and treatment of therapy-induced adverse effects. Electronic patient reported outcome measures (ePROMs) like mobile and Webb applications (apps) offer the opportunity to address particular unmet needs of cancer patients. They can be utilised for screening, diagnostic, therapeutic and educational purposes [3]. Various in-app questionnaires facilitate capturing patient-reported outcomes, like symptom burden, physical function, mental status and quality of life [4–8]. Implementing electronic monitoring of treatment side effects into routine breast cancer care decreases chemotherapy-induced symptom burden, and improves symptom management, self-efficacy and quality of life [8–13]. Moreover, several studies exploring mobile apps encouraging physical activity among breast cancer patients demonstrated improved quality of life and reduced fatigue and distress levels [14–16]. Additionally, results of research testing eHealth solutions during the COVID-19 pandemic not only confirmed that remote monitoring and management of treatment-related symptoms reduces symptom burden and improves quality of life, but also limits unplanned healthcare utilisation, decreasing demand on healthcare systems and improves patients’ symptom experience, including perception of the frequency, intensity, distress, and meaning of symptoms [6]. Furthermore, eHealth solutions have been identified as empowering, improving patients involvement in the continuum of cancer care [17,18].

The primary objective of this study was to analyse the results of using an in-app electronic questionnaire to assess and monitor chemotherapy-related symptoms in patients treated for early-stage breast cancer.

The aim of the study is clearly defined.

Thank you for this comment.

In the Methodology section inclusion criteria should be clarified, indicating if all types of breast cancer were included in the current study.

Corrected to:

Patients who met the following inclusion criteria were eligible for participation in this study: referral for perioperative chemotherapy for early-stage breast cancer (all types), possession of their own smart device, ability to navigate the device and download mobile applications independently, and completed a signed informed consent form. Patients with metastatic breast cancer and those treated for other types of cancers were excluded from the study. 

Also the medication used by patients, in addition to chemotherapy, should have been considered.

Thank you for your suggestion, however this was not considered to be a relevant factor at the stage of planning our study, thus this information was not collected. The intention was to collect information in the setting of “real-world” treatment, including routine use of wide range of concomitant medications, including those used to treat toxicities of chemotherapy and our aim was to assess the added value of collection of information on toxicities, not to evaluate the impact of concomitant medications.

For further clarification, we have added ‘Details of concomitant medication were not collected for the purpose of this study’.

The economic status classification should be clarified, regarding the tool used to this assessment by self report (I believe this is the way it was done).

Thank you for this suggestion. Corrected to:

Nursing staff collected medical data (cancer stage and phenotype, setting of chemotherapy (preoperative vs. postoperative), type of cytotoxic drugs used, number of cycles, coexisting medical conditions and use of granulocyte colony-stimulating factors) from patients’ health records. Sociodemographic details (age, sex, place of residence, education, employment, economic and marital status) were self-reported by participants at the end of the patient satisfaction survey.

Methods for statistical analysis are appropriate to the study design purposed.

Thank you for your valued comment.

In the Results section I leave the suggestion of split Table 1 in two tables, one describing sociodemographic characteristics and another presenting clinical data (including cytotoxic drugs).

Thank you for this comment. We have separated data in Table 1 as suggested.

Table 1. Sociodemographic characteristics of study participants

Table 2. Medical characteristics of study participants.

Also Fig.3 can be replaced by another figure, table or even descriptive text, it does not add any relevant information.

Thank you for your suggestion, however authors would prefer to keep this figure to present true distribution of collected responses, including also the single participant who visibly exaggerated number of reports.

In the graph of Fig.4 it would be advisable for each bar to have the value of the respective frequency.

Thank you for this suggestion. It has been corrected.

Table 2 is long and difficult to read, I suggest it be rearranged or included as supplementary material. Furthermore, there are no comments in the text regarding the contents of this table.

Thank you for your valued comment. We have moved Table 2 to supplementary material as suggested.

Information provided through Fig.7 is not easily understandable, this information includes the answers for questions Q1-Q8?? Please clarify the information provided in the legend or change the graph used.

Thank you for this suggestion. We decided to remove fig.7 completely and replace it by tab. 3.

Later, questions presented in appendix A, the question number is not correspondent to those used in the Fig.7.

Thank you for this comment. Appendix A has been corrected.

The Conclusions could be reformulated in order to present the study main achievements more clearly, since it does not address exactly the information passed in the results section.

Thank you for your valued comment. We have changed this part as suggested:

Although successful use of ePROMs for monitoring treatment-related adverse events has been described in many settings, the results of this study suggest possible lack of trust and/or understanding of eHealth tools among Polish patients treated for early-stage breast cancer. Our findings suggest that patients older than 60 years of age find it difficult to engage with mobile technology and eHealth solutions. On the other hand, this is the population that, according to our research, is at higher risk of experiencing not only cancer-related fatigue, but also other problems caused by the treatment. To improve patient engagement and understanding of eHealth solutions, it is essential that patients are invited and involved in fundamental stages of creating innovative app-based interventions. With age being a significant factor in determining the number of problems experienced during chemotherapy, we suggest that additional support is provided to older patients to enhance their awareness of the beneficiary potential of eHealth interventions. More research involving older participants is needed to explore and address their particular needs and perspectives on eHealth solutions.

The references are appropriate, up to date (more than 80% of the references were published in the last 10 years).

Thank you for this comment.

In general, the figures and graphs should be improved regarding their presentation.

Thank you for this comment, we have made suggested changes.

Comments on the Quality of English Language

In general, writing should be revised throughout the entire manuscript.

Thank you for you comment. The entire manuscript has been revised by native English speaker.

Reviewer 2 Report

Peer review report for the manuscript “Not all societies ready for digital tools. Feasibility study on the use of mobile application in Polish early breast cancer patients treated with perioperative chemotherapy”

The aim of this manuscript was to analyze the results of using in app electronic questionnaire to assess and monitor chemotherapy-related symptoms in patients treated for early-stage breast cancer. Current and very interesting topic.

I have some comments.

11. Abstract: There is not information about why the topic is important for readers; there is no justification for the seriousness of the problem

22. Material and method: The section was prepared clearly and in detail. I only suggest to:

a)     remove information about anthracycline-taxane regimens – this information (line 59-64) is not important for the topic.

b)    in line 59 words “table 1” used mistakenly. Figure 1 should be corrected  (there is information in Polish)

33. Results:

a)     what were the criteria for inclusion and exclusion of patients from the study?

b)    in table 1 information about chemotherapy regimen is not necessary.

c)     number of tables should be reviewed and corrected (e.g. in text we have figure 6a-d, and next A, B, C, D.

d)    after this text “Figure 7 presents proportion of responses collected in the patient’s satisfaction survey” (line 211-112) authors should explain what mean Q1-Q8. It is inconvenient to look for this information in a supplement

44. Discussion: You wrote: “High acceptance of digital tools in cancer care is widely recognized” – please add some references;

Limitations of research and alternatives are missing

55. Conclusion: what will be recommend for other research and practice?

Author Response

Peer review report for the manuscript “Not all societies ready for digital tools. Feasibility study on the use of mobile application in Polish early breast cancer patients treated with perioperative chemotherapy”

The aim of this manuscript was to analyze the results of using in app electronic questionnaire to assess and monitor chemotherapy-related symptoms in patients treated for early-stage breast cancer. Current and very interesting topic.

I have some comments.

  1. Abstract: There is not information about why the topic is important for readers; there is no justification for the seriousness of the problem

Thank you for this comment. Following text has been added to the abstract:

Background: The population of individuals affected by breast cancer is growing, and with advances in cancer treatment implemented into usual care, there is an urgent need to improve recognition, monitoring and treatment of therapy-induced adverse effects. This study aims to explore the use of an in-app electronic questionnaire to assess and monitor of chemotherapy-related symptoms in early breast cancer patients treated with perioperative chemotherapy.

  1. Material and method: The section was prepared clearly and in detail. I only suggest to:
  2. a)     remove information about anthracycline-taxane regimens – this information (line 59-64) is not important for the topic.

Thank you for this suggestion, however treatment regimen was considered a possible factor influencing number of reports and communicated problems, hence detailed explanation of specific regimens was provided for readers.

  1. b)    in line 59 words “table 1” used mistakenly.

Thank you for this comment, however considering previous response, authors felt that readers would like to see what treatment regimens were used during the study. 

 Figure 1 should be corrected  (there is information in Polish)

Thank you for this comment, however this a true screenshot of Centrum Chorób Piersi UCK mobile app, which is in Polish. We have adjusted the figure by adding English translations below Polish words. Logotypes including Centrum Chorób Piersi and UCK Uniwersyteckie Centrum Kliniczne Efektywnie leczÄ…c, dobrze uczyć i sÅ‚użyć nauce should remain in Polish. Referring to another Reviewer’s suggestion, figure 1 has been moved  between text:

… The application did not allow omitted questions or free-text responses. Reports generated in connection to patients' in-app activity were closely monitored by the BCN. Figure 1 demonstrates a simplified patient – breast care team communication process via the in-app questionnaire.

Figure 1

Replies including symptoms with critical value (≥3, apart from fever, which activated alerts when rated as 1 or above) triggered automatic email alerts to the nursing team. Participants were informed that after triggering an alert they will be contacted by the BCN within one working day during office hours.…

  1. Results:
  2. a)     what were the criteria for inclusion and exclusion of patients from the study?

Corrected to:

Patients who met the following inclusion criteria were eligible for participation in this study: referral for perioperative chemotherapy for early-stage breast cancer (all types), possession of their own smart device, ability to navigate the device and download mobile applications independently, and completed a signed informed consent form. Patients with metastatic breast cancer and those treated for other types of cancers were excluded from the study. 

  1. b)    in table 1 information about chemotherapy regimen is not necessary.

Thank you for this suggestion, however treatment regimen was considered a possible factor influencing number of reports and communicated problems, hence detailed explanation of specific regimens was provided for readers.

  1. c)     number of tables should be reviewed and corrected (e.g. in text we have figure 6a-d, and next A, B, C, D.

Thank you for this comment. We have made appropriate changes.

  1. d)    after this text “Figure 7 presents proportion of responses collected in the patient’s satisfaction survey” (line 211-112) authors should explain what mean Q1-Q8. It is inconvenient to look for this information in a supplement

Thank you for this comment. We decided to remove fig.7 completely and replace it by tab. 3.  The table presents detailed responses collected in patient’s satisfaction survey.

  1. Discussion: You wrote: “High acceptance of digital tools in cancer care is widely recognized” – please add some references;

Thank you for this suggestion. It has been updated.

Limitations of research and alternatives are missing

Thank you for your valued comment. Following study limitations have been added:

The present study has some limitations. Our project was performed in a single centre with a limited number of participants due to the COVID-19 pandemic. To explore participants' perception of the app, we have used a proprietary survey, instead of a standardised tool, however, we have achieved a 100% completion rate with overall positive feedback. Another potential drawback is that the study design did not involve automatic reminders for participants to complete the in-app questionnaire, resulting in low app adherence (e.g., only 6% of participants completed the questionnaire as instructed).

  1. Conclusion: what will be recommend for other research and practice?

Thank you for this comment. Referring also to another review, we have added (from line 399):

To improve patient engagement and understanding of eHealth solutions, it is essential that patients are invited and involved in fundamental stages of creating innovative app-based interventions. With age being a significant factor in determining the number of problems experienced during chemotherapy, we suggest that additional support is provided to older patients to enhance their awareness of the beneficiary potential of eHealth interventions. More research involving older participants is needed to explore and address their particular needs and perspectives on eHealth solutions.

Reviewer 3 Report

The study question is interesting and the content is well presented. An app to support breast cancer patients during chemotherapy is very useful. I have some minor points:

1) Unfortunately, the interest of the participants decreased after the study. How do the authors intend to counteract this? Are there any ideas on how to do this?

2) Were medical support contacted in the year after the study?

3) It would also be interesting to know the medical staff's experience with the additional burden of the calls. Have the staff been asked about this? Please discuss whether this call burden would be possible in standard care.

4) Table 2 is very confusing. Perhaps the authors could use different shades of gray to separate the individual columns. 

5) Appendix A. Patient’s satisfaction Survey: Could the authors please provide correct enumeration of the Questions. 

6) Table S1. Detailed symptom assessment scale: Could the authors please explain why some topics have 5 possible answers and some have only 4?

Thanks in advance.

Author Response

The study question is interesting and the content is well presented. An app to support breast cancer patients during chemotherapy is very useful. I have some minor points:

1) Unfortunately, the interest of the participants decreased after the study. How do the authors intend to counteract this? Are there any ideas on how to do this?

Authors will consider liaising with app designers to modify the functionality of the app allowing patient’s reminders to improve app adherence. Advertising campaign promoting the app will be repeated among patients of our Breast Cancer Unit.

2) Were medical support contacted in the year after the study?

It is a standard procedure for breast care nurses to monitor responses and alerts collected via this app. Moreover, all patients, including study participants, have open access to contact breast nurses when necessary. However, as this was during a period outside our study, we did not review the data on these contacts. Moreover, it was not included in the approval of Ethics Committee of Medical University of Gdańsk.

3) It would also be interesting to know the medical staff's experience with the additional burden of the calls. Have the staff been asked about this? Please discuss whether this call burden would be possible in standard care.

Thank you for the comment. Staff perspective was not officially considered within this study. Nevertheless, the Breast Cancer Chemotherapy Unit where our study was conducted has well established breast specialist nursing care. Telephone consultations are primary ways of communication. Considering that our study took place during peak of the COVID-19 pandemic, telephone consultations were already fundamental way of patient – breast care team communication and calls resulting from mobile app-generated reports did not pose significant additional burden on our staff. However, in view of growing demand on patient support and limited amount of available, competent staff, one can expect that e-health solutions will become indispensable in the routine care of breast cancer patients.

4) Table 2 is very confusing. Perhaps the authors could use different shades of gray to separate the individual columns. 

Thank you for your suggestion. In response to other Reviewers’ suggestions, table 2 has been moved to supplementary materials.

5) Appendix A. Patient’s satisfaction Survey: Could the authors please provide correct enumeration of the Questions. 

Thank you for this comment. It has been corrected.

6) Table S1. Detailed symptom assessment scale: Could the authors please explain why some topics have 5 possible answers and some have only 4?

Symptom assessment scale used in our study is based on Common Terminology Criteria for Adverse Events v.4 with only minor alternations. Number of responses referring to specific symptoms is consistent with the original scale.

Reviewer 4 Report

An interesting paper about a wonderful topic, such as  the use of mobile application in Polish early breast cancer patients  treated with perioperative chemotherapy and its feasibility. I would like to make some comments and suggestions about the paper:

- The abstract has got a lot of valid information about the study. I recommend using the structure Intro (it doesn´t exist in this abstract)- Method-Results-Conclusions, to highlight the different parts of the research. 

- The introduction is too brief and it doesn´t give enough information about the topic. Please, write again this part with more and better information about the topic. 

- About the Methodology, I also recommend splitting the parts in Procedure-Measures-Sample, in order to clarify the explanation. 

- Figure 1 is not well explained and referred into the text (and, in my opinion, it doesn´t give any scientific information). 

- About the survey, it would be great yo add the validity and feasibility of the data collected. 

- Results are wonderful, only I would like to suggest presenting the data from Table 2 in another way: it´s a little bit visually complicated.

- There is no part of limitation in the paper, I consider it fundamental for publication (for example, the number of participants avoids any kind of external validity of the study). Also, there is no measure of the influence by lockdown or any other external/strange variable in the study

Author Response

An interesting paper about a wonderful topic, such as  the use of mobile application in Polish early breast cancer patients  treated with perioperative chemotherapy and its feasibility. I would like to make some comments and suggestions about the paper:

- The abstract has got a lot of valid information about the study. I recommend using the structure Intro (it doesn´t exist in this abstract)- Method-Results-Conclusions, to highlight the different parts of the research. 

Thank you for this valued comment. We have adjusted the abstract as suggested.

- The introduction is too brief and it doesn´t give enough information about the topic. Please, write again this part with more and better information about the topic. 

Thank you for your valued comment. Referring also to other Reviewers, we have changed the introduction as follows:

Breast cancer is the most prevalent cancer diagnosis and the second most frequent cause of cancer-related deaths among Polish women. According to the Polish National Cancer Registry in 2020, 23.8% of primary cancer diagnoses in women were breast cancers. While the population of people diagnosed with breast cancer in Poland is growing [1] and advances in cancer treatment are being implemented into usual care [2], there is an urgent need to improve the recognition, monitoring and treatment of therapy-induced adverse effects. Electronic patient reported outcome measures (ePROMs) like mobile and Webb applications (apps) offer the opportunity to address particular unmet needs of cancer patients. They can be utilised for screening, diagnostic, therapeutic and educational purposes [3]. Various in-app questionnaires facilitate capturing patient-reported outcomes, like symptom burden, physical function, mental status and quality of life [4–8]. Implementing electronic monitoring of treatment side effects into routine breast cancer care decreases chemotherapy-induced symptom burden, and improves symptom management, self-efficacy and quality of life [8–13]. Moreover, several studies exploring mobile apps encouraging physical activity among breast cancer patients demonstrated improved quality of life and reduced fatigue and distress levels [14–16]. Additionally, results of research testing eHealth solutions during the COVID-19 pandemic not only confirmed that remote monitoring and management of treatment-related symptoms reduces symptom burden and improves quality of life, but also limits unplanned healthcare utilisation, decreasing demand on healthcare systems and improves patients’ symptom experience, including perception of the frequency, intensity, distress, and meaning of symptoms [6]. Furthermore, eHealth solutions have been identified as empowering, improving patients involvement in the continuum of cancer care [17,18].

The primary objective of this study was to analyse the results of using an in-app electronic questionnaire to assess and monitor chemotherapy-related symptoms in patients treated for early-stage breast cancer.

- About the Methodology, I also recommend splitting the parts in Procedure-Measures-Sample, in order to clarify the explanation. 

Thank you for this suggestion. To clarify methodology, we have split this part into: 2.1 Study design and participants, 2.2. Measures, 2.3. Data analysis.

- Figure 1 is not well explained and referred into the text (and, in my opinion, it doesn´t give any scientific information). 

Thank you for this comment. We have adjusted the figure by adding English translations below Polish words. Logotypes including Centrum Chorób Piersi and UCK Uniwersyteckie Centrum Kliniczne Efektywnie leczÄ…c, dobrze uczyć i sÅ‚użyć nauce should remain in Polish. The figure has been moved  between text:

… The application did not allow omitted questions or free-text responses. Reports generated in connection to patients' in-app activity were closely monitored by the BCN. Figure 1 demonstrates a simplified patient – breast care team communication process via the in-app questionnaire.

Figure 1

Replies including symptoms with critical value (≥3, apart from fever, which activated alerts when rated as 1 or above) triggered automatic email alerts to the nursing team. Participants were informed that after triggering an alert they will be contacted by the BCN within one working day during office hours.…

- About the survey, it would be great to add the validity and feasibility of the data collected. 

The app explored in our study has never been tested before, therefore we have no data for comparison. Information about treatment side effects collected via the electronic questionnaire was assesses with standardised tool, however patient satisfaction was described by participants on proprietary survey questionnaire. We believe that both measures are reliable and results presented in our study are reproducible.

- Results are wonderful, only I would like to suggest presenting the data from Table 2 in another way: it´s a little bit visually complicated.

Thank you for your valued comment. In response to other Reviewers’ suggestions, table 2 has been moved to supplementary materials.

- There is no part of limitation in the paper, I consider it fundamental for publication (for example, the number of participants avoids any kind of external validity of the study). Also, there is no measure of the influence by lockdown or any other external/strange variable in the study

Thank you for your valued comment. Following study limitations have been added:

The present study has some limitations. Our project was performed in a single centre with a limited number of participants due to the COVID-19 pandemic. To explore participants' perception of the app, we have used a proprietary survey, instead of a standardised tool, however, we have achieved a 100% completion rate with overall positive feedback. Another potential drawback is that the study design did not involve automatic reminders for participants to complete the in-app questionnaire, resulting in low app adherence (e.g., only 6% of participants completed the questionnaire as instructed).

Round 2

Reviewer 2 Report

Thank you very much for the revision of the manuscript. The authors have done a nice job revising this paper. I have one comment for part "recommening for other research and practice". You wrote that "we suggest that additional support is provided to older patients to enhance their awareness of the beneficiary potential of eHealth interventions". In my opinion that support is nessesery for all patients. Your post-study observations could be evidence. 

Reviewer 4 Report

Lot of thanks for your efforts. There is only one lack in the paper, and it´s the absence of validity and feasibility data on the survey. The rest of the work is really well improved.